# Synthesis of Multifunctional Eu(III) Complex Doped Fe_3_O_4_/Au Nanocomposite for Dual Photo-Magnetic Hyperthermia and Fluorescence Bioimaging

**DOI:** 10.3390/molecules28020749

**Published:** 2023-01-11

**Authors:** Hoang Thi Khuyen, Tran Thu Huong, Nguyen Duc Van, Nguyen Thanh Huong, Nguyen Vu, Pham Thi Lien, Pham Hong Nam, Vu Xuan Nghia

**Affiliations:** 1Institute of Materials Science, Vietnam Academy of Science and Technology, 18 Hoang Quoc Viet, Cau Giay, Hanoi 100000, Vietnam; 2Department of Materials Science and Energy, Graduate University of Science and Technology, Vietnam Academy of Science and Technology, 18 Hoang Quoc Viet, Cau Giay, Hanoi 100000, Vietnam; 3108 Military Central Hospital, 01 Tran Hung Dao, Hai Ba Trung, Hanoi 100000, Vietnam

**Keywords:** nanocomposite, hyperthermia, imaging, Eu(TTA)_3_, Fe_3_O_4_, Au

## Abstract

In this paper, the luminescent complex Eu(3-thenoyltrifluoroacetonate)_3_ was integrated with Fe_3_O_4_ and gold (Au) nanoparticles to form a multifunctional nanocomposite, Fe_3_O_4_/Au/Eu(TTA)_3_ (FOASET NC), for dual magnetic-photothermal therapy and biomedical imaging. Upon functionalization with amine-NH_2_, the FOASET NC exhibits a small size of 60–70 nm and strong, sharp emission at λ_max_ = 614 nm, enhanced by surface plasmon resonance (SPR) of Au nanoparticles that provided an effective label for HT29 colorectal cancer cells by fluorescence microscopy imaging. In addition, a hyperthermia temperature (42–46 °C) was completely achieved by using these FOASET NCs in an aqueous solution with three heating modes for (i) Magnetic therapy (MT), (ii) Photothermal therapy (PT), and (iii) Dual magnetic-photothermal therapy (MPT). The heating efficiency was improved in the dual magnetic-photothermal heating mode.

## 1. Introduction

Recently, multifunctional nanostructures have played an important role in developing therapeutic agents, such as drug delivery, magnetic therapy, and photothermal therapy, that could be combined with biomedical imaging [1,2,3]. The surface functionalized and bioconjugated nanomaterials will bring a specific targeted therapy and diagnosis [1,4]. Nanoparticles (NPs)-based hyperthermia has received significant attention as a promising approach in cancer treatment without painful surgical intervention, highly toxic radiotherapy, or chemotherapy [5,6]. Magnetic hyperthermia is a useful technique developed to treat cancer cells at (42–46 °C) using magnetic nanoparticles (MNPs) under the influence of an alternating magnetic field (AMF). Among magnetic materials for hyperthermia applications, much attention has been focused on iron oxide nanoparticles (such as Fe_2_O_3_ and Fe_3_O_4_) due to their small size, good dispersion in solution, low toxicity, chemical stability, and special superparamagnetic behavior [7,8]. Besides, plasmon nanoparticles, nanotubes of gold, silver, etc. have also been used as photoabsorbers to convert optical energy into heat to kill cancer cells [3,5,9,10].

However, a significant challenge for the development and hyperthermia applications of these NPs in the clinic is to improve the heating efficiency with an appropriate dose, temperature, and duration [11]. Accordingly, most research has been focused on the effects of size, shape, and concentration of MNPs, as well as physical parameters such as frequency and amplitude of the AMF, on the heating efficiency [12,13]. More recently, one of the main aims for NPs-based hyperthermia is to synthesize multifunctional nanoparticles that exhibit high heat generation efficiency and have appropriately functionalized surfaces to bioconjugate and selectively attach to target cells. Some functional materials, such as silica, polymers, plasmon noble metals (gold, silver, etc.), and luminescent materials, have been coated and combined with magnetic nanomaterials to improve the heating efficiency, improve biocompatibility, and expand their applicability towards theranostics [14,15,16,17,18,19,20].

Biomedical fluorescence image based on luminescent materials is used to detect viruses and cells as well as estimating their biodistribution in both in-vitro and vivo [21,22]. Some luminescent materials, such as organic dyes, semiconductor quantum dots (QDs), and luminescent materials containing rare earth ions, have been extensively studied [14,23,24,25,26]. Several organic dyes have been commercialized to identify ions or molecules associated with viruses and cells. However, they have some disadvantages, such as poor photochemical stability, broad emission spectra, high photobleaching, and autofluorescence [23]. The rare earth ion-based nanomaterials exhibit a large Stokes shift, narrow photoluminescent spectra, high quantum yields, and long lifetimes that limit the noise of spontaneous or background photoluminescence. However, the inorganic rare earth nanomaterials are only excited by ultraviolet (UV) light, which can affect cells. The rare earth up-conversion nanomaterials are capable of absorbing near-infrared (NIR) light with high penetration, but their luminescence is usually weak. Nanomaterials containing europium(III) complexes with organic ligands have received a great deal of attention due to their broad visible (VIS) light sensitized excitation spectra and their luminescence enhanced by appropriate organic ligands. Besides, they are more photochemically stable than organic dyes and quite environmentally friendly [27,28]. The integration of magnetic and plasmonic nanoparticles with luminescent materials in unique multifunctional nanosystems enables the creation of combined therapeutic and bioimaging tools with multi-functionality, multi-mode, and controlled biodistribution. This is an important area of research.

Based on the abovementioned considerations, a new multifunctional FOASET NC based on the luminescent Eu(3-thenoyltrifluoroacetonate)_3_ complex (Eu(TTA)_3_) has been synthesized for dual-targeted hyperthermia and fluorescence bioimaging applications. The characterization and magnetic–optical properties of the FOASET nanoparticles were investigated. By using these multifunctional FOASET nanoparticles, an in vitro test for labelling HT29 colorectal cancer cells was carried out. Their dual magnetic-photothermal heating generation efficiency for hyperthermia was also evaluated.

## 2. Results and Discussion

### 2.1. Characterization and Properties of the Multifunctional FOASET NC

Figure 1 presents photo images of the as-synthesized FOASET NC. Observed from the experiment, the FOASET NC is composed of black colloidal nanoparticles with good dispersion in an aqueous solution (Figure 1a). When the FOASET NC solution cuvette was placed near a magnet (under the influence of a magnetic field), the FOASET nanoparticles were quickly attracted to the direction of the magnet, as indicated in Figure 1b. When the sample solution cuvette was placed near a magnet and excited by light, the FOASET nanoparticles were attracted to the magnet and emitted a red luminescence simultaneously (Figure 1c). The results demonstrated that the synthesized multifunctional FOASET NC are able to possess both magnetic property and red luminescence.

The nanoparticle size of less than 100 nm is perfectly suitable for biomedical applications. Figure 2a,b, and c show a TEM image of the Fe_3_O_4_ nanoparticles and SEM images of the Fe_3_O_4_/Au (FOA) and FOASET nanoparticles, respectively. The FOASET NC was synthesized from Fe_3_O_4_ nanoparticles with a small size of around 5–10 nm, as indicated by their SEM image. The FOA nanoparticles have an average size of 35–40 nm. After being coated with an amine-functionalized silica layer doped with a luminescent Eu(TTA)_3_ complex, the FOASET nanoparticles are spherical with an average size of 60–70 nm.

The size distribution and zeta potential of these colloidal nanoparticles were studied by dynamic light scattering, as shown in Figure 3. The hydrodynamic sizes of the Fe_3_O_4,_ Fe_3_O_4_/Au (FOA), and FOASET nanoparticles were 27.50, 55.07, and 168.2, respectively (Figure 3a). The FOASET NC was functionalized by TESPA to improve their biocompatibility and facilitate the integration of the luminescent Eu(TTA)_3_ complex. The silica coating also provides electrostatically stabilized nanoparticles and reduces agglomeration. A zeta potential value of about −30 mV was measured (Figure 3b). Compared with FOA nanoparticles (Appendix A), the colloidal stability of FOASET nanoparticles was improved. Figure 4a,b, and c show the XRD patterns of Fe_3_O_4_, FOA, and FOASET nanoparticles, respectively. All the detectable peaks of the curve a, at 2θ = 30.51° 36.03°, 43.41°, 54,36°, 57.97°, and 63.08°, were contributed by the lattice surfaces of (220), (311), (400), (422), (511), and (440), respectively, which correspond to the cubic Fe_3_O_4_ crystal structure (according to ICSD using POWD –12 ++95, 561 (1915)). On XRD patterns of FOA and FOASET NC, the peaks at 2θ = 38.38°, 44.08°, and 65.76°, respectively, were related to the (111), (200), and (220) lattice surfaces of a cubic fcc crystal structure of Au nanoparticles (according to Swanson, Tatge., Natl. Bur. Stand. (US), Circ. 539 I, 33 (1953)) (curves b and c). In addition, an amorphous phase of the amine-NH_2_ functionalized silica layer has also formed by the appearance of a broad, diffuse halo in 2θ of 15°–30° in the XRD diagram of FOASET NC (curve c). This result is consistent with the EDX elemental analysis of the FOASET NC.

The main elemental composition of the multifunctional FOASET NC was analyzed by EDX measurement (Figure 5). The Fe, O, Au, Eu, C, and S elements appeared corresponding to a contribution of the nanoparticles of Fe_3_O_4_, Au, and Eu(TTA)_3_ complex. The Si, O, C, and N elements were from the formation of an amine-NH_2_-functionalized silica layer. No impurities were found.

Figure 6a–c shows the FTIR spectra of the Fe_3_O_4,_ FOA, and FOASET nanoparticles. All samples show an absorption peak at a low vibration frequency of about 580 cm^−1^, which corresponds to the characteristic Fe–O bond. A broad peak at 3330 cm^−1^ is related to a vibration of the O–H bond. The N–H bond is also observed in this band. Figure 6b,c show the peaks at approximately 1042 and 1083 cm^−1^, which are characteristic peaks of the Si–O–R group, respectively. The peaks at 1635 and 1451 cm^−1^ appeared in the FTIR spectrum of FOASET nanoparticles, corresponding to the stretching vibrations of the C=C bond, C=O (ketone group), and C–F (organic TTA ligand), respectively (Figure 6c). The peaks at 1290 cm^−1^ are related to the P=O bond of TOPO. The aromatic =C–H stretching was also observed at 3100–3000 cm^−1^. The Eu–O bond is contributed by vibrations in the low-frequency region of 500–1000 cm^−1^.

The magnetic hysteresis loop of the FOASET nanoparticles at room temperature in the applied magnetic field in the range of +100,000 to −10,000 Oe was shown in Figure 7. The FOASET nanoparticles show a superparamagnetic property with narrow hysteresis loops and very small amounts of residual magnetism. At the applied magnetic field of ±10,000 Oe, a magnetization of 36.41 (emu/g) was obtained. It should be noted that full saturation magnetization is not reached at H ± 10,000 Oe.

Figure 8 exhibits the UV–VIS spectra of FOA and FOASET nanoparticles, respectively. From UV–Vis absorption spectra, the FOA NC shows an SPR peak at 531 nm that was attributed to Au nanoparticles (curve a). The multifunctional FOASET NC exhibits a broad absorption band extending from the UV–VIS to the NIR region. In curve b, two peaks at 262 nm and 345 nm are related to the absorption characteristics of the Eu(TTA)_3_ complex. They correspond to intramolecular charge transfer (ICT) of TTA ligands and a TTA ligand to europium(III) charge transfer, respectively. The broad absorption band from 400 nm to 900 nm induced by SPR of the Au nanoparticles layer with a peak at 528 nm was also obviously observed. The broad optical absorption of the synthesized multifunctional FOASET nanoparticles allows them to overcome the limitation of UV excitation of rare earth ions and makes them potentially useful for photothermal therapy.

In Figure 9A, the curves of (a) and (b), respectively, present the luminescent spectra of the FOASET nanoparticles and Eu(TTA)_3_ complex under excitation at λ_exc_ = 355 nm at room temperature. They show a red emission of the europium(III) ion with a main transition (^5^D_0_ → ^7^F_2_) λ_max_ = 614 nm. Their luminescent mechanism was associated with an indirect excitation of the europium(III) ion (an antenna effect). The organic TTA ligand absorbs excitation energy and transfers this energy to the triplet state. From here, the energy is transferred to the ^5^D_j_ levels, and a luminescent emission occurs through the transition from ^5^D_0_ to the levels of the ground state of ^7^F_j_ (j = 0, 1, 2, 3, and 4). The luminescence of multifunctional FOASET nanoparticles is stronger than that of the Eu(TTA)_3_ complex at the same concentration. The luminescent intensity of the FOASET nanoparticles was enhanced by a factor of 3.31 times. This was induced by the SPR of Au nanoparticles, leading to a local electromagnetic field enhancement. Figure 9B shows the luminescent spectra of the FOASET nanoparticles under excitation at λ_exc_ = 532 nm. The results indicate that the integration with Au nanoparticles extended the absorption band toward the visible wavelength, leading to an efficient excitation at λ_exc_ = 532 nm. The broad absorption band from UV to NIR range, red strong luminescence induced by SPR of Au nanoparticles, and good luminescent stability are important advantages of the multifunctional FOASET NC for a biomedical imaging application.

### 2.2. In Vitro Fluorescence Bioimaging Test of HT29 Colorectal Cancer Cells

In this study, the in vitro test for labelling HT29 colorectal cancer cells was carried out. The HT29 colorectal cancer cells were implanted and maintained in the Eagle’s Minimum Essential Medium (EMEM). The integration of the Eu(TTA)_3_ complex in the multifunctional FOASET nanostructure allows for achieving biomedical fluorescence imaging. As mentioned above, the multifunctional FOASET nanoparticles were functionalized with the amine-NH_2_ group. Therefore, GDA was used as a biological bridge between the FOASET nanoparticles and a specific IgG antibody, forming a FOASET–IgG nanocomplex. The HT29 colorectal cancer cells were incubated with the FOASET–IgG nanocomplex for immunofluorescence identification.

The fluorescence images of the HT29 colorectal cancer cells in the bright field (BF) and dark field (DF) modes were shown in Figure 10. A sample of HT29 colorectal cells without the FOASET–IgG nanocomplex was used as a comparison sample. Figure 10a,c, respectively, show the images of the HT29 colorectal cells without/with the FOASET–IgG nanocomplex in the BF mode. In the DF mode, the samples were excited by light. The viability of cells was evaluated by direct observation of cells under an inverted phase contrast microscope. From the results, the cells have good viability after being incubated with the FOASET–IgG nanocomplex. After careful washing, the cells adhered well to the bottom of the culture plate. The images of the HT29 colorectal cancer cells with the FOASET–IgG nanocomplex were observed clearly in both BF and DF modes (Figure 10c,d, respectively). Figure 10b,d show the image of the HT29 colorectal cells in the DF mode. As observed in Figure 10b, a fluorescence image of the HT29 colorectal cells sample without the FOASET–IgG nanocomplex was not obtained. Meanwhile, the red fluorescence image of the HT29 colorectal cells incubated with the FOASET–IgG nanocomplex in the well was clearly observed in this mode (Figure 10d). The FOASET–IgG nanocomplex was attached to HT29 colorectal cells through an antigen–antibody combination. The result clearly indicates that the synthesized FOASET–IgG nanocomplex had a good attachment to the HT29 colorectal cancer cells, therefore leading to efficient identification of the HT29 colorectal cells by fluorescence microscopy imaging.

### 2.3. Heating Generation Efficiency for Hyperthermia

Figure 11A shows the heating curves of the FOASET nanoparticles. The FOASET nanoparticles were dispersed in deoxygenated water at a concentration of 2 mg mL^−1^. Three heating generation modes, including MT, PT, and MPT modes, were studied. The MT mode was carried out by using an alternating magnetic field at a frequency of 340 kHz and an amplitude of 200 Oe for 800 s. For the PT mode, the colloidal FOASET nanoparticles were irradiated by a laser at 808 nm with a power density of 0.25 W cm^−2^. For the MPT mode, both MT and PT modes were applicated simultaneously. 

From the experimental results, a heating generation efficiency was obtained in all three modes. A temperature of 42–46 °C that is required in hyperthermia [1] was achieved by using the multifunctional FOASET nanoparticles in MT, PT, and PMT modes. After a short period of 800 s, these temperatures were 43.5, 42.3, and 53.2 °C, respectively. It is worth noting that a hyperthermia temperature was found in the MPT mode after only about 136 s.

The heating efficiency of the FOASET nanoparticles is characterized by a specific absorption rate, SAR, which can be experimentally determined using the following equation:$$\mathrm{SAR}=\frac{C}{m}\frac{\mathrm{dT}}{\mathrm{dt}}\;\left(\frac{W}{g}\right)$$

C is the specific heat capacity of water (4.185 J g^−1^ K^−1^), m is the concentration (mg mL^−1^) of the FOASET nanoparticles in the solution, and dT/dt is the slope of the measured heating curve [18]. 

Table 1 presents the experimental values of SAR and dT/dt of the FOASET NC. The SAR values of the FOASET NCs in MT, PT, and MPT modes are 118.80, 136.48, and 200.25 W. g^1^, respectively (Figure 11B). Compared with the MT and MP modes, the SAR value of the multifunctional FOASET NCs has increased in the MPT modes. A synergistic SAR value was achieved by using the MPT model with laser irradiation at power (0.25 W.cm^−2^) and under the influence of the alternating magnetic field (200 Oe, 340 kHz). Therefore, the heating generation efficiency has been improved in the dual magnetic-photothermal heating mode (MPT mode). A synergistic heating generation in MPT mode that was based on the magnetic-photothermal properties of the FOASET NC, allowing control and reduction of the amplitude of the alternating magnetic field, power density of laser, and time for hyperthermia application. The synthesized FOASET NC possesses the superparamagnetic property and SPR absorption induced by Au nanoparticles. Therefore, the MT, PT, and MPT modes can be selected and used to achieve suitable and efficient heating generation for the targeted hyperthermia.

## 3. Materials and Methods

### 3.1. Materials

Gold (III) chloride trihydrate (HAuCl_4_·3H_2_O), tri-n-octylphosphineoxide (TOPO), 3-thenoyltrifluoroacetonate (TTA), 3-(triethoxysilyl)-propylamine (TESPA), ammonium hydroxide (NH_4_OH), ascorbic acid (AA), and glutaraldehyde (GDA) were purchased from Aldrich. Europium(III) chloride hexahydrate (EuCl_3_.6H_2_O), and hexadecyltrimethylammonium bromide (CTAB) were purchased from Sigma Aldrich (Burlington, MA, USA). Sodium dodecyl sulfate (SDS) and ethanol were from Merck (Darmstadt, Germany). Fe_3_O_4_ nanoparticles with a size of about 10 nm were synthesized in our laboratory of Photochem, Imaging and Photonics, IMS, and VAST.

### 3.2. Synthesis of the Multifunctional FOASET NC

The multifunctional FOASET NC was synthesized by a modified Stober method according to our previous work [26]. This simple wet chemical synthesis route was carried out at room temperature, enabling efficient control of particle shape and size and being convenient for functionalization and bioconjugation. The steps of the synthesis of the multifunctional FOASET NC were illustrated in the Appendix A. Briefly, a mixture of 0.025 mM HAuCl_4_, 0.01 M CTAB, and 0.1 M AA was added twice to a three-neck round-bottom flask containing Fe_3_O_4_ nanoparticles. Then, 50 µL TESPA was added and pH was adjusted to 8.5 by NH_4_OH solution. The sample was stirred for 3 h to obtain an Au-coated Fe_3_O_4_ nanocomposite (FOA NC). Then, it was washed and transferred to an ethanol solvent. The Eu (TTA)_3_ complex was prepared from a mixture of EuCl_3_.6H_2_O, TTA, and TOPO in the mole ratio of 1:3:2.6. The 5 mM Eu (TTA)_3_ complex and 0.1 mM SDS were introduced to the FOA NC solution. Then, 500 µL TESPA was added and stirred for 12 h. After a washing process, the FOASET NC was dispersed in an aqueous solution and dried at 60 °C for 24 h.

The HT29 colorectal cells were from HTB-38, ATCC, USA. The HT29 colorectal cells were put into a 6-well plate (at 10^4^ cells/well) and incubated at 37 °C, 5% CO, for 24 h. The cells were then fixed with 10% formaldehyde for 10 min at room temperature. The formaldehyde was then removed, and the cells were washed three times with PBS. In addition, 10 µL of the FOASET–IgG nanocomplex in PBS solution was added to each well and further incubated at 4 °C for 1 h. Then, the sample was washed several times with PBS before the cells were observed under fluorescence microscopy.

### 3.3. Methods

The morphology of the as-synthesized FOASET NC was observed on a scanning electron microscope (SEM, Hitachi-SU8100, Tokyo, Japan). Zeta potential and size distribution were determined by dynamic light scattering (DLS), using a Zetasizer Nano ZS, London, UK. The crystal phase identification was characterized by X-ray diffraction (XRD, Equinox 5000, Paris, France). The FTIR spectra were analyzed using a Nexus 670 ThermoNicolet Fourier transform infrared spectrometer, Waltham, MA, USA. The UV–VIS absorption spectra were recorded at room temperature by a UV-VIS Biochrom s60, London, UK instrument system. The fluorescence spectra of the FOASET NC were measured in a quart cuvette with a 1 cm part length with an IHR 550 Hobira Jobin Yvon spectrometer (Waltham, MA, USA). Magnetic measurement was studied using a vibration sample magnetometer (VSM, Hanoi, Vietnam). The fluorescence cell image was observed by a Nikon Eclipse Ti2 fluorescence microscope, Tokyo, Japan. The heating generation with magnetic-thermal, photothermal, and dual magnetic-photothermal modes of the FOASET NC was measured by using a commercially available generator (UHF-20A, Waltham, MA, USA) at a magnetic field strength of 200 Oe at a frequency of 340 kHz combined with laser irradiation (LD808E3WG13) at a wavelength of 808 nm with a power density of 0.25 W.cm^−2^.

## 4. Conclusions

In this work, the multifunctional nanocomposite, Fe_3_O_4_/Au/Eu(TTA)_3_ (FOASET NC), was synthesized by a simple modified Stober method. The FOASET nanoparticles exhibit a small size of 60–70 nm, good dispersion, and a broad absorption band induced by the SPR of Au nanoparticles. Their strong red luminescence was able to be excited by UV light at λ_exc_ = 355 nm and VIS light at λ_exc_ = 532 nm. The FOASET nanoparticles functionalized with amine-NH_2_ were successfully used for the in vitro fluorescence image to identify the HT29 colorectal cancer cells. In addition, the FOASET nanoparticles demonstrated a dual synergistic magnetic and photothermal heating generation, leading to an improved heating efficiency for hyperthermia therapy. The multifunctional FOASET NC provided a promising candidate for targeted hyperthermia therapy and biomedical applications.

## Figures and Tables

**Figure 1 molecules-28-00749-f001:**
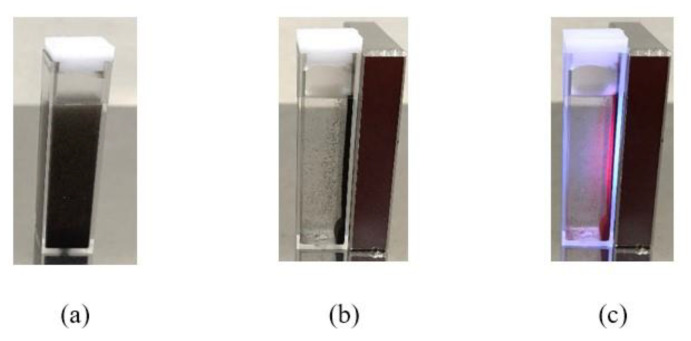
Photo images of FOASET NC, (**a**) at ambient temperature, (**b**) under the influence of a magnet, and (**c**) under the influence of magnet and light excitation.

**Figure 2 molecules-28-00749-f002:**
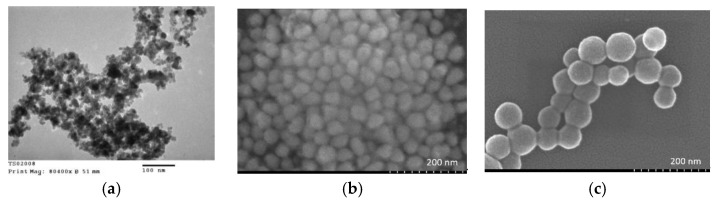
(**a**) TEM images of Fe_3_O_4_ nanoparticles and SEM images of (**b**) FOA and (**c**) FOASET nanoparticles.

**Figure 3 molecules-28-00749-f003:**
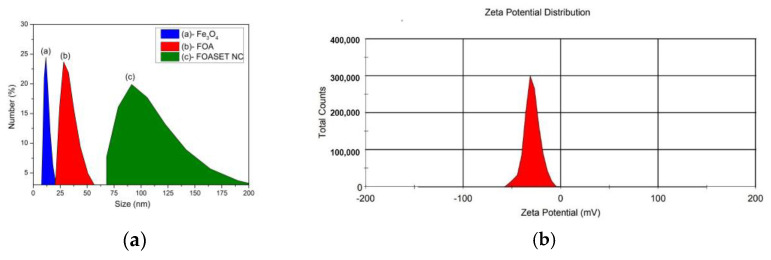
(**a**) Size distribution of Fe_3_O_4_, FOA, and FOASET nanoparticles, and (**b**) zeta potential of FOASET nanoparticles.

**Figure 4 molecules-28-00749-f004:**
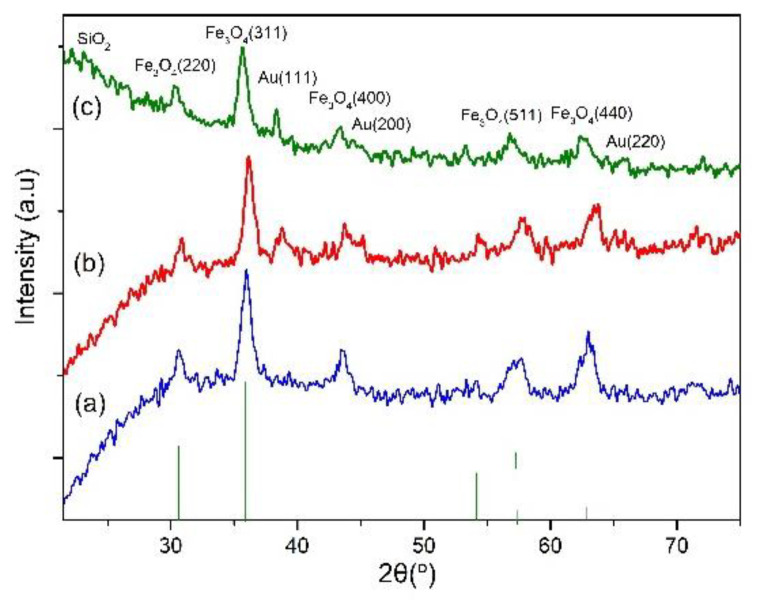
XRD patterns of (**a**) Fe_3_O_4_, (**b**) FOA, and (**c**) FOASET nanoparticles.

**Figure 5 molecules-28-00749-f005:**
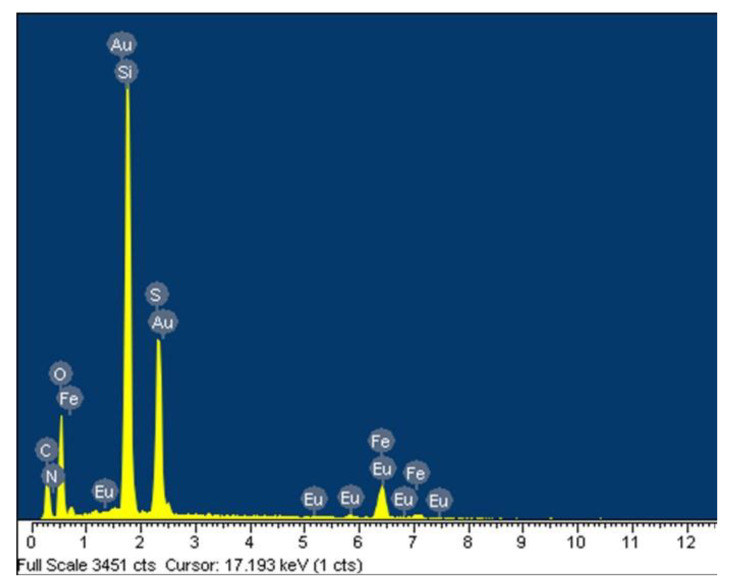
EDX elemental analysis of FOASET NC.

**Figure 6 molecules-28-00749-f006:**
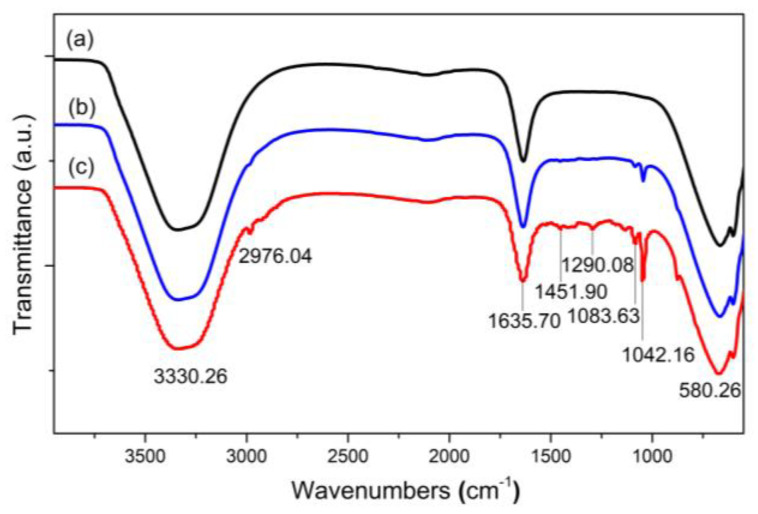
FTIR spectra of (**a**) Fe_3_O_4_, (**b**) FOA, and (**c**) FOASET nanoparticles.

**Figure 7 molecules-28-00749-f007:**
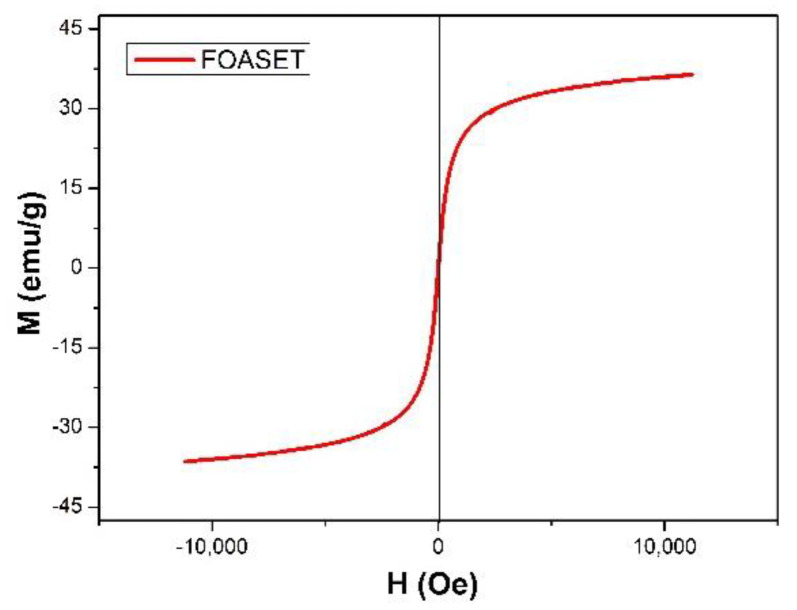
Magnetic hysteresis loops of FOASET nanoparticles.

**Figure 8 molecules-28-00749-f008:**
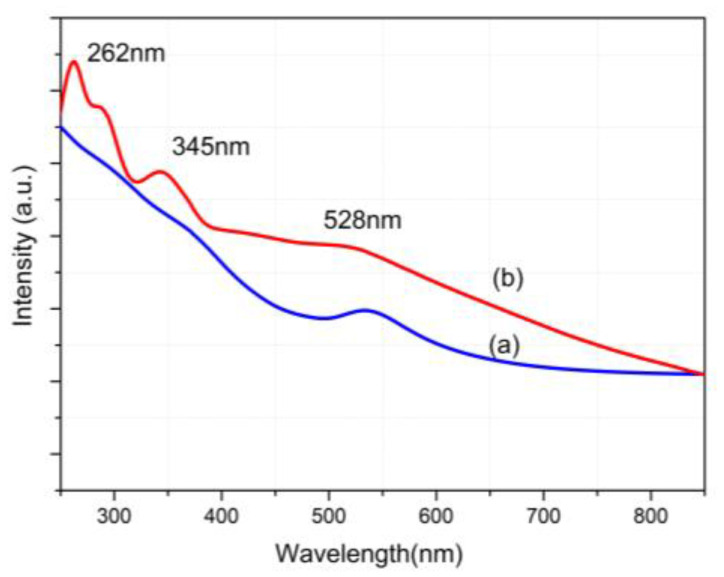
UV–VIS spectra of (**a**) FOA and (**b**) FOASET nanoparticles.

**Figure 9 molecules-28-00749-f009:**
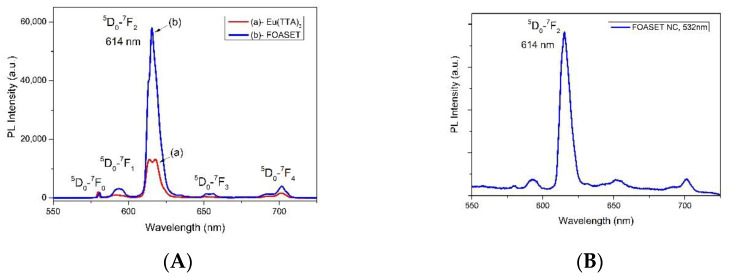
(**A**) Luminescent spectra of (a) FOASET nanoparticles and (b) Eu(TTA)_3_ complex under excitation at λ_exc_ = 355 nm and (**B**) at λ_exc_ = 532 nm.

**Figure 10 molecules-28-00749-f010:**
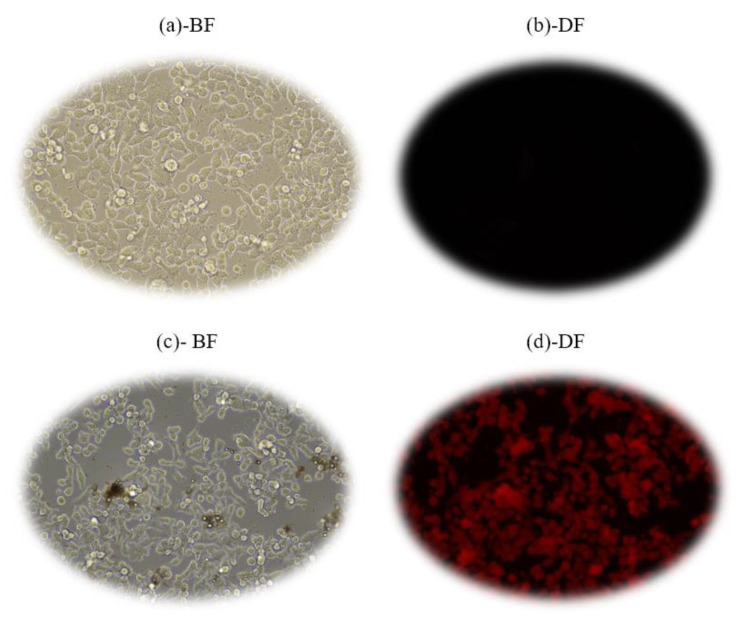
Fluorescent image of the HT29 colorectal cells without FOASET-IgG nanoparticles (**a**)in BF and (**b**) in DF, and with FOAET-IgG nanoparticles (**c**) in BF and (**d**) in DF.

**Figure 11 molecules-28-00749-f011:**
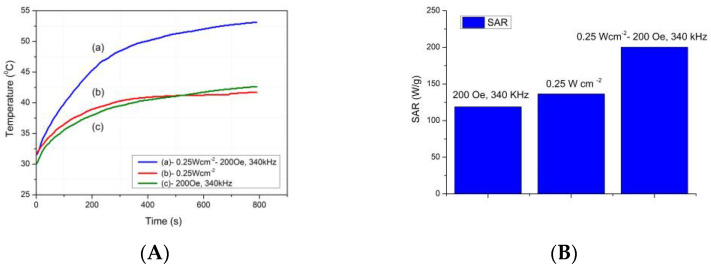
(**A**) Heating curves of the FOASET nanoparticles with (a) MT, (b) PT, and (c) MPT modes; (**B**) SAR calculations of the FOASET nanoparticles with different heating modes.

**Table 1 molecules-28-00749-t001:** Experimental values of SAR and (dT/dt) of FOASET NC.

Sample	H, f (Oe, kHz)	P (W/cm^2^)	dT/dt (°C/s)	SAR (W/g)
FOASET NC	200 Oe, 340 kHz		0.054	118.80
		0.25 W/cm^2^	0.062	136.48
	200 Oe, 340 kHz	0.25 W/cm^2^	0.091	200.25

## Data Availability

Data are available via personal communication with proper reasons.

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
