# Peer review of "Synthesis of Multifunctional Eu(III) Complex Doped Fe3O4/Au Nanocomposite for Dual Photo-Magnetic Hyperthermia and Fluorescence Bioimaging"

_molecules, 2023, doi:10.3390/molecules28020749_

Round 1

Reviewer 1 Report

In the manuscript entitled " Synthesis of multifunctional Eu(III) complex doped Fe3O4/Au nanocomposite for dual photo-magnetic hyperthermia and fluorescence bioimaging" (Manuscript Number: molecules-2147451), the authors propose that Eu(3-thenoyltrifluoroacetonate)3 was integrated with Fe3O4 and gold (Au) nanoparticles to form a multifunctional nanocomposite, Fe3O4/Au/Eu(TTA)3 (FOASET NC) for dual magnetic-photothermal therapy and biomedical imaging. This manuscript might be an interesting topic for publication, there are several points that require more careful examination. I have major and minor comments exhibited below.

Major comments:

1.      Please add DLS data to show the average size of Fe3O4, Fe3O4/Au (FOA) and FOASET nanoparticles;

2.      In the manuscript write “Fe3O4 nanoparticles with a small size of around 5-10 nm”, please add Low- and high-magnification TEM images;

3.      In order to prove that it is Fe3O4 nanoparticles, it is required to do the X-ray diffraction (XRD) pattern;

4.      In the manuscript write the FOASET nanoparticles have biological imaging performance and should be tested in terms of biocompatibility, but I haven't seen any relevant description, should add relevant experiments.

5.      In the introduction part, should add more related reference about photo-magnetic hyperthermia such as Nanomaterials 12 (8), 1370, Journal of Materials Chemistry B 10 (34), 6532-6545, Journal of Materials Chemistry B 8 (36), 8356-8367.

Minor comments:

1.      SEM images of Fe3O4 need to change a clear one.

2.      Replace the image of Zeta potential with a column or other form of image to see the results more clearly.

3.      In the Figure. 10, I haven’t see the error bar.

4.      In the line 200, For the PT mode, the colloidal FOASET nanoparticles were irradiated by a laser at 808 nm with a power density of 0.25 W cm-2. Why did you choose 0.25 W cm-2 such low power density?

Author Response

Dear Prof., 

First of all, we wish you and your journal an excellent year 2023! We would like to thank you for your valuable comments. These comments led us to an improvement of the manuscript. The “point-to-point” responses to your comments are listed below. We believe that the revised manuscript is at appropriate level for publication in “Molecules”. 

Your sincerely,

Hoang Thi Khuyen

Reviewer 2 Report

In this manuscript, the authors designed a new multifunctional nanocomposite (FOASET NC), consisting of Fe3O4, gold (Au) nanoparticles, and Eu(3-thenoyltrifluoroacetonate)3 for dual magnetic-photothermal therapy and biomedical imaging. However, the authors must clarify the following points before publishing this work in Molecules:

1. The authors are suggested to provide a scheme to show the synthesis of FOASET NC step by step.

2. In Figure 3, the authors should also measure the zeta potential of FOA nanoparticles to compare with FOASET nanoparticles.

3. In Figure 8, the authors compared the luminescent spectra of FOASET nanoparticles and Eu(TTA)3 complex. Did the authors control the Eu contents of these two types of nanoparticles for luminescence measurement to be similar?

Author Response

Dear Prof., 

First of all, we wish you and your journal an excellent year 2023! We would like to thank you for their valuable comments. These comments led us to an improvement of the manuscript. The “point-to-point” responses to your comments are listed below. We believe that the revised manuscript is at appropriate level for publication in “Molecules”. 

Your sincerely,

Hoang Thi Khuyen

Reviewer 3 Report

Nomenclature should be clarified - e.g. are FOASET NC the same as  Fe3O4 nanoparticles? Cell culture experiment should be described in the Materials and Methods section including information where HT29 cells were purchased. The cells were not implanted (L165) but they were cultured in EMEM.

Author Response

(The authors gave the same response as above.)

Round 2

Reviewer 1 Report

plz, pay attention to the quality of the figures. Thanks